# Impact of Dilution on Whisky Aroma: A Sensory and Volatile Composition Analysis

**DOI:** 10.3390/foods12061276

**Published:** 2023-03-17

**Authors:** P. Layton Ashmore, Aubrey DuBois, Elizabeth Tomasino, James F. Harbertson, Thomas S. Collins

**Affiliations:** 1Department of Food Science, Washington State University, Richland, WA 99354, USA; 2Department of Food Science and Human Nutrition, Michigan State University, East Lansing, MI 48824, USA; 3Department of Food Science & Technology, Oregon State University, Corvallis, OR 97331, USA; 4Department of Viticulture and Enology, Washington State University, Richland, WA 99354, USA

**Keywords:** whisky, aroma, descriptive analysis, HS-SPME-GC-MS, volatile composition

## Abstract

An “omics”-style approach was used to evaluate the complex relationship between whisky aroma and dilution with water, typically suggested as a way to better appreciate whisky. A set of 25 samples, including Bourbons, ryes, single-malt and blended Scotches, and Irish whiskies were chemically profiled at six dilution levels (100, 90, 80, 70, 60, and 50% whisky/water), while a subset of six whiskies (three Bourbons, three Scotches) at four dilution levels (100, 80, 60, and 40% whisky/water) were chemically profiled and subjected to sensory analysis by a trained panel (*n* = 20). Untargeted volatile analysis was performed using headspace solid-phase microextraction gas chromatography coupled with mass spectrometry (HS-SPME-GC-MS) and sensory analysis was performed using descriptive analysis (DA). Results were evaluated using multivariate statistical techniques, including multifactor analysis (MFA) and partial least squares discriminant analysis (PLS-DA). Dilution decreased headspace concentration of hydrophilic aroma compounds and increased concentration of more hydrophobic compounds, which agreed with DA results. Dilution above 80% whisky/20% water reduced differences within whisky styles, though differences between American (Bourbon, rye) and Scotch styles (single malt, blended) continued to increase with further dilution. This provides important insight into how dilution of whisky during consumption changes consumer perception, as well as the usefulness of HS-SPME-GC-MS as a proxy for human olfaction.

## 1. Introduction

Conventional wisdom suggests that the best way to experience a glass of whisky is to add water to “open” the whisky aroma. On a molecular level, Henry’s Law dictates that the partial pressures of gasses in the headspace above a liquid mixture are directly proportional to their concentration in the mixture itself. At first glance, this would suggest that any dilution with water would likewise dilute headspace concentration of any aroma compounds, making them less noticeable. However, this behavior becomes more complicated when accounting for hydrophobicity of said compounds and the matrix composition in which they exist. This is better understood by examining the Henry’s Law constant, also known as the air-water partitioning coefficient (K_aw_), which is a function of the compound in question [1]. Similarly, the hydrophobicity of a compound is often calculated as its octanol-water coefficient (K_ow_), usually expressed as log(K_ow_) or logP, which is an indication of a compound’s preference to solubilize in a more polar matrix (water) versus a more nonpolar matrix (n-octanol) [2]. Further complicating the matter is the complexity of human olfaction, where a given “aroma” is often due to a mixture of multiple compounds [3,4] and detection thresholds differ depending on matrix composition [5] as well as which other types of aroma compounds are present in the headspace [6].

Recent attempts to model this partitioning behavior in silico suggest that amphipathic molecules such as 2-methoxyphenol (guaiacol) migrate to the liquid-air interface layers upon dilution with water, where they are more likely to escape into the headspace, enhancing their aroma [7]. While guaiacol and other volatile phenols are characteristic compounds associated with barrel aging [8], it represents only a small portion of overall whisky aroma and does not represent how headspace concentration of more hydrophobic or hydrophilic compounds changes upon dilution with water, nor does it capture the complex nature of interactions between aroma compounds and how they are experienced by the olfactory system.

Overall whisky aroma is derived from each step in the production process, including raw ingredients, fermentation, distillation, and barrel aging. Aroma components derived from raw materials are generally characteristic of stylistic choice. For example, peated malts used in many styles of Scotch whisky impart the classic smokey aromas associated with such styles [9] while high corn content is responsible for the milder characteristics associated with American Bourbons [10]. While inclusions of other grains such as rye and wheat can impart unique flavors to a final product, their impact on overall aroma is less clear [11]. Aromas associated with fermentation are generally related to production of short and medium-chain fatty acids and their respective ethyl esters, as well as ethanol and higher, fusel alcohols [12]. Distillation factors, such as legal requirements for fresh whisky proof, result in differing concentrations of congeners, or non-alcohol aroma compounds. Higher congener concentration results in more characteristics from raw materials and fermentation [13], and lower congener concentration, or alternatively higher initial whisky proof, results in more extraction of barrel components, resulting in a product with more characteristics related to toasted or charred wood [14]. Barrel aging imparts aroma via varying levels of volatile phenols due to thermal breakdown of lignin polymers during the toasting and charring process [15]. These levels are, in turn, affected by barreling proof [14], level of toasting/charring [15], wood type and physical properties [16], whether the barrel has been previously used [17], and how long the whisky is exposed [18], and the conditions of the barrel room during aging [19].

Using a single compound to model such a diverse population of aroma active species is insufficient and represents a major limitation of in silico modeling for such dynamic systems. While calculating the air-water partitioning coefficient of each individual component of every style of whisky would give a more encompassing model, this is impractical if not impossible due to the complexity of the whisky matrix. Instead, a strategy utilizing untargeted volatile profiling via headspace solid phase microextraction coupled with mass spectrometry (HS-SPME-GC-MS) and sensory descriptive analysis (DA) to elucidate a more complete picture of how whisky aroma behaves under dilution with water is proposed. By gaining better understanding of the totality of aroma interactions of various whisky styles and how they behave under dilution, a common phenomenon when whisky is enjoyed with ice or a “splash” of water, producers can gain important insight to how their product is perceived, not just initially but throughout the entire consumption. This work can also provide important insight into how effective HS-SPME-GC-MS is as a proxy for human olfaction.

## 2. Materials and Methods

### 2.1. Reagents and Glassware

Class-A volumetric glassware was used for all analytical sample preparations. Reagents and solvents were ACS reagent grade or higher. Deuterated (d_8_) nonanone (>98% deuterated) was chosen as an internal standard and purchased from CDN Isotopes (Pointe-Claire, Quebec, QC, Canada). Dilutions were prepared using ultrapure (18.2 MΩ·cm) water (MilliporeSigma, Burlington, MA, USA). Whiskies were purchased from local retailers.

### 2.2. Sample Preparation

A total of 25 whiskies were chosen for volatile profiling: six Bourbons, six single malt Scotches, five blended Scotches, four Irish whiskeys, and four rye whiskeys (Appendix A). A smaller selection of six whiskies (three Bourbons, three Scotches) of the same alcohol content (43% *v*/*v*) were chosen for sensory analysis as well as volatile profiling.

Sample preparation and subsequent profiling were performed according to a previously published method with minor modifications [20]. Samples were prepared for chemometric profiling in 50-mL volumetric flasks where 120 µg/L of internal standard (d_8_-nonanone, ISTD) and appropriate volume of purified water were added by micropipette (Eppendorf Research Plus, Hamburg, Germany). Six dilution levels (100, 90, 80, 70, 60, 50% *v*/*v* whisky in water) were prepared and aliquots of 10 mL were placed in to triplicate 20-mL glass screw cap headspace vials (Restek, Bellefonte, PA, USA). Whiskies used in descriptive analysis panels were prepared with 120 µg/L ISTD at dilution levels of 100, 80, 60, and 40% *v*/*v* whisky using ultrapure water (MilliporeSigma, Burlington, MA, USA) and likewise analyzed in triplicate.

### 2.3. Volatile Profiling

Untargeted volatile profiling of whiskies was performed using headspace solid-phase microextraction gas chromatography with mass spectrometry (HS-SPME-GC-MS) with an Agilent 6890/5975 system (Agilent Technologies, Santa Clara, CA, USA) and a Gerstel Robotic MPS autosampler (Gerstel, Inc., Linthicum, MD, USA). Samples were allowed to equilibrate at 30 °C for 1 min then extracted for 40 min using a Supelco Stableflex 50/30 µm divinylbenzene/carboxen/polydimethylsiloxane fiber (Supelco, Inc., Bellefonte, PA, USA) with a vial penetration depth of 21 mm and an agitator speed of 250 rpm. The SPME fiber was desorbed in the inlet for 2 min at 250 °C, the inlet split was reduced to 5:1 from the published method for a moderate increase in sensitivity. A polar column (DB-WAX-UI, 30 m × 0.250 mm ID, 0.25 µm film thickness, Agilent Technologies, Santa Clara, CA, USA) was used for separations. The GC oven was initially held at 40 °C for 3 min, followed by a temperature ramp of 3.5 °C/min to 180 °C with no hold and a final temperature ramp of 30 °C/min to 250 °C with a 3 min hold. The MSD transfer line was held at 260 °C and fragmentation data were collected for masses between 40 and 300 *m*/*z* with a source temperature of 230 °C, quadrupole temperature of 150 °C, and electron impact energy of 70 eV. The detector was turned off from 3.5 to 4.5 min to allow for the solvent peak. Total sample run time was 48 min/injection.

### 2.4. Sensory Analysis

Six 86-proof commercial whiskies (three Bourbons and three Scotches) were selected for sensory profile analyses. Each whisky was diluted with purified distilled water to 80%, 60%, and 40% (*v*/*v*) whisky and an undiluted preparation of each served as controls. Sensory testing was approved by the Oregon State University Internal Review Board (Study #8606) and took place in the Arbuthnot Dairy Center at Oregon State University (Corvallis, OR, USA). White tri-fold poster boards were used to create individual testing booths. Each booth was equipped with a Chromebook to access Compusense Cloud (v21.0.7713.26683, Compusense Inc., Guelph, ON, Canada) survey software. The testing environment was kept at ambient temperatures under a mixture of natural and incandescent lighting.

Lexicon selection was determined through pilot testing and a check-all-that-apply (CATA) evaluation using untrained consumers. Terms with the highest frequency were selected for use in a general descriptive analysis (DA) panel. All participants were at least 21 years of age and drank liquors such as whiskey at least once a week. Potential subjects were excluded from the study if they were currently pregnant, a smoker, had food allergies, taste or smell deficits, active oral sores or lesions, or piercings of the tongue, lip, or cheek. 25 subjects (16 female, 8 male, 1 gender-neutral) participated in the CATA test and 20 (12 female, 7 male, 1 gender-neutral) participated in the descriptive analysis. After the DA panel, samples of whiskies were taken for volatile profiling described in Section 2.3.

#### 2.4.1. Check-All-That-Apply (CATA)

Fifteen aroma attributes were selected based on preliminary pilot testing of the samples: bacon, band-aid, brine, cedar, cornmeal/cooked polenta, hay, malt, oak, peat smoke, pine, pome fruit, rubber, sawdust, solvent/chemical, and vanilla. A complete block design was used to present the 24 samples over two sessions. Attribute presentation order within each question was randomized to mitigate attribute order biases. 12 samples were presented per session, with enforced one-minute breaks between each sample and three-minute breaks between each flight of four whiskies. All samples were served in black Glencairn glasses (Glencairn Crystal Studio, East Kilbride, UK) with plastic watch glasses (Choice Foodservice Products, Layton, UT, USA) placed over top and labeled with a random three-digit code.

Panelists were asked to smell each sample and check all attributes they felt best described the aroma of the sample. In addition to the 15 aroma attributes, two additional attributes labeled “other” were included such that panelists could add an attribute not listed if they felt it was important to the sample’s aroma.

#### 2.4.2. Descriptive Analysis (DA)

Ten attributes with the most significant differences between stimuli from the CATA results were selected for descriptive analysis: band-aid, cedar, cornmeal/cooked polenta, malt, oak, peat smoke, pome fruit, rubber, solvent/chemical, and vanilla. A 10-cm visual analog scale (VAS) anchored with “none” at 0.3 cm and “extreme” at 9.7 cm was used to rate aroma intensities of the samples.

Prior to sample evaluation panelists were trained over two sessions to recognize the attributes of interest and to familiarize themselves with use of the VAS. Panelists were presented with attribute standards (Appendix A) and instructed to smell each standard sample and then select which aroma term described the aroma. Standards and attributes were presented to panelists using randomized complete block designs. After evaluation of the standards, panelists were presented with two whiskies (chosen at random), presented monadically, to practice using the VAS to rate attribute intensities. 1-min breaks were enforced between samples and a 3-min break was enforced after six to prevent sensory fatigue.

Following training, participants evaluated the 24 samples over four sessions. A randomized complete block design was used to determine presentation order and each panelist evaluated all samples twice. Subjects were given 20-mL scintillation vials of each aroma standard for reference throughout each session. Twelve samples were evaluated per session, with enforced one-minute breaks between samples and an enforced three-minute break between flights of six. For each sample, panelists were asked to rate the intensity of each of the terms using the VAS.

### 2.5. Data Processing and Statistical Analysis

Simultaneous chromatogram deconvolution of HS-SPME-GC-MS data was performed using non-negative PARAFAC2 modeling software (PARADISe v5.9, University of Copenhagen, Copenhagen, Denmark) [21]. Tentative compound identifications were acquired using NIST MS Search (v2.4, National Institute of Standards and Technology, Gaithersburg, MD, USA) using the NIST20 library. Statistical analyses were performed in R (v4.2.1) using RStudio (v2021.09.0 build 351, RStudio, Boston, MA, USA) and the FactoMineR, factoextra, and mixOmics packages. For the larger dataset of 25 assorted whiskies, 91 volatile compounds were tentatively identified and separated in to eight compound classes (Appendix A). For the smaller dataset of whiskies used for sensory analysis, 131 volatile compounds were tentatively identified and similarly separated in to 11 compound classes (Appendix A). Compound classes were used for multifactor analysis (MFA) in R. These classes were roughly chosen according to the types of compounds present. For example, ethyl esters are generally produced during fermentation by *Saccharomyces cerevisiae* [22] and were subsequently grouped together, as were PAHs and benzofurans which notably stem from peat smoke exposure during the malting process in Scotch whisky production [23].

For sensory analysis, CATA data were analyzed via Cochran’s Q test to determine significant differences between stimuli for each attribute. Ten total attributes were chosen to use in the subsequent DA panel, “Bacon”, “Cedar”, “Cornmeal/cooked polenta”, “Malt”, “Oak”, “Peat smoke”, “Pome fruit”, “Rubber”, “Solvent/Chemical”, and “Vanilla”. Overall significance of attributes as determined by the DA panel was evaluated by MANOVA, and individual ANOVAs were used to determine significance of individual descriptors and their interactions. A mixed effects model was used to determine which descriptors with significant sample:subject and sample:session interactions remained significant.

## 3. Results

### 3.1. Untargeted Profiling of Twenty-Five Assorted Whiskies

The MFA performed on the larger whisky dataset using previously described compound classes is shown in Figure 1.

A total of 24.7% of the variance is represented in the first two principle components (PC1 = 12.9% and PC2 = 11.8%). Confidence ellipses were distinct (non-overlapping) for 100, 90, and 80% whisky dilution levels. Starting at the 70% whisky dilution level differences became less distinct, with confidence ellipses overlapping between 70 and 60% whisky as well as 60 and 50% whisky. Whisky styles were significantly different, with principle component 1 separating blended Scotch (BS) and single malt Scotch (SM) from Bourbon (BU) and principle component 2 separating rye whiskey (RW) and Bourbon (BU) from barrel-strength Bourbon (BB) and Irish whiskey (IW). Individual samples (whisky brand) show some differences and similarities based on 95% confidence ellipses.

### 3.2. Untargeted Profiling and Sensory Analysis of Six-Whisky Subset

#### 3.2.1. Sensory Panel Performance and Results

Results of the CATA evaluation can be found in the Appendix A. Of the initial 15 attributes used to describe the whisky dilution series, ten (“Bacon”, “Cedar”, “Cornmeal/Cooked polenta”, “Malt”, “Oak”, “Peat smoke”, “Pome fruit”, “Rubber”, “Solvent/Chemical”, and “Vanilla”) were selected for use in further evaluation.

Evaluation of DA panel performance determined seven of the ten descriptors used (“Vanilla”, “Solvent/Chemical”, “Pome fruit”, “Bacon”, “Rubber”, and “Peat smoke”) showed significant differences (*p* < 0.05) between treatments (Appendix A). Panelists were able to evaluate these seven descriptors in a consistent manner.

#### 3.2.2. Untargeted Profiling of Sensory Whiskies

An MFA, performed using the compound classes described above, is shown in Figure 2.

A total of 51.8% of variance was accounted for in the first two principal components (PC1 = 27%, PC2 = 24.8%). The 100 and 80% whisky/water dilution levels were significantly different while 60 and 40% whisky/water dilutions were not. All samples displayed significant differences with the exception of two Bourbons. All three styles displayed significant differences at each dilution level.

Partial least squares (PLS) analysis was used to determine significant correlations between chemical compounds and associated descriptors (Figure 3). Number codes and their associated compounds can be found in the Appendix A.

Of the ten descriptors, seven had significant associations with chemical profiling (“Pome fruit”, “Oak”, “Vanilla”, “Solvent/Chemical”, “Bacon”, “Rubber”, and “Peat smoke”) while three did not (“Cornmeal/polenta”, “Cedar”, and “Malt”). This mostly agrees with the significant descriptors as determined by DA panel with the exception of the “Oak” attribute.

In general, the “acetate esters” class of compounds was commonly associated with “Pome fruit” aroma descriptors while the “phenols” class was more commonly associated with the smokey attributes “Bacon”, “Rubber”, and “Peat smoke”. This is more easily seen in their respective correlograms (Figure 4A,B). Correlograms for the remaining compound classes can be found in the Appendix A.

Partial least squares discriminant analyses (PLS-DA) were used to determine overall associations between whisky style and sensory descriptors used by the trained panel and by dilution factor (Figure 5).

Smoke-related attributes of “Rubber”, “Bacon” and “Peat smoke” were highly predictive of peated single malt Scotch, while the descriptors of “Oak” and “Vanilla” were strong predictors of Bourbons. “Pome fruit” was a strong predictor of blended Scotch whisky. In general, undiluted whiskies were more associated with “Chemical/Solvent”, “Vanilla”, “Oak”, and “Bacon” aromas while diluted whiskies were associated with “Pome Fruit”, and “Cedar” aromas.

## 4. Discussion

The HS-SPME-GC-MS method was able to effectively integrate and identify 87 compounds in the large (25 whisky) dataset and 131 compounds in the smaller (6 whisky) dataset. The simultaneous deconvolution by PARADISe provided a convenient and effective method for both data alignment and automated identification of compounds by NIST library lookup. The lower number of identified compounds in the larger dataset was due to difficulty identifying individual peaks in such a large number of overlapping chromatograms. High ethanol concentration of the undiluted whisky samples was a concern for SPME analysis, as this can lead to fiber competition from ethanol overwhelming the concentration of volatiles of interest [24,25]. However, sensory studies in wine have suggested that higher ethanol concentrations produce a masking effect of many aroma-active compounds [26,27]. Due to the much higher concentration of ethanol in whisky as compared to wine, it would be expected that not only would this also occur in whisky, but to a much larger extent. Therefore, fiber competition by ethanol can be seen as a comparable process to the overwhelming of olfactory receptors by ethanol masking individual aroma compounds in lower concentrations. This can be observed to an extent in Figure 5B where “solvent/chemical” aroma is strongly associated with undiluted whiskies and negatively correlated with dilution level. More specifically, as noted by le Berre, et al. (2007), “woody” characteristics tend to be enhanced with ethanol concentration while other aromas are masked, which is also seen in Figure 5B, as “vanilla” and “oak” are barrel-related aroma descriptors and are positively correlated with ethanol concentration [28].

Overall effects of whisky dilution with water on aroma are best exemplified in the PLS-DA analyses of the sensory data (Figure 5). Bourbon and rye whiskies, which by law are aged in first-use charred American oak, were heavily associated with “Oak” and “Vanilla” aromas, which are common descriptors for some of the most aroma-active compounds typically found in Bourbons (volatile phenols) [10]. As mentioned previously, these phenols are introduced through the oak charring process, where thermal degradation of lignin results in the release of phenol monomers [15]. The decrease in “Vanilla” aroma with dilution shown in Figure 5B would be expected due to the amphipathic nature of phenols such as vanillin and guaiacol (logP values around 1, [2]), as well as general dilution effects. Alternatively, “Oak” aroma appears to evolve into a “Cedar” aroma with dilution, possibly due to the hydrophobic nature of many of the oak lactones found in Bourbon which are often described as “raw wood” or “pencil shavings” [29]. “Pome fruit” was associated with single malt Scotch style whiskies and acetate esters. These esters are derived from acetic acid and fusel alcohols, which were also notably higher in single malt Scotch style whiskies and tend to provide “fruity” aromas [22]. They are also relatively hydrophobic compounds with logP typically greater than 2 [2], which suggests they are more likely to be repelled by water into the headspace above the sample. The evolution of aromas with dilution is also likely a factor of certain constituents of the overall aroma falling below recognition and/or detection thresholds as the sample matrices are changed [5].

The MFAs of the two datasets (Figure 1 and Figure 2) showed a notable loss in significant differences in whisky clustering around the 80% whisky/20% water dilution level. The fact that both datasets behaved similarly despite containing different whiskies, different dilution levels, and being built using different compounds/classes suggests that, in general, this dilution level is where headspace concentrations of volatiles become less distinct, resulting in a loss of noticeable differences between individual whiskies. Therefore, further dilution can be seen as having a deleterious effect on whisky aroma. However, it is worth noting that while the trends upon further dilution suggest that while telling individual whiskies within a style apart may become more difficult, the differences between American styles of whiskey (Bourbon and Rye) become increasingly different from Scotch (blended and single malt) and Irish styles (Figure 1).

The strong PLS correlations between DA panel responses and chemical composition of whiskies seen in Figure 3, as well as the associations shown in PLS-DA plots in Figure 5, suggest a strong level of predictability in sensory attributes from untargeted chemical profiling. Previous work has shown that near IR spectral data can be used as a predictor of sensory attributes in foods [30] as can targeted chemical analysis of grapes can predict sensory attributes of wines [31,32]. While prediction of sensory attributes was not the goal of this study, the predictive ability of untargeted volatile profiles of whiskies and their dilutions shows promise. Being able to take a finished product and quickly determine not only how it will be perceived by a consumer, but also how that perception will change upon dilution with water or a melting ice cube would be a valuable tool for producers.

## 5. Conclusions

Untargeted volatile profiling of a range of whisky styles at various dilution levels has shown to be a valuable tool to determine how whisky is experienced by consumers. By coupling with trained sensory panel analysis, the behavior of the whisky “volatilome” has shown strong differences according to whisky style as well as individual whisky samples. Furthermore, dilution past the 80% whisky/20% water level has shown to be deleterious to whisky aroma, resulting in diminishing ability to distinguish between whiskies. Further work examining the predictability of sensory attributes from volatile profiles shows promise, due to strong correlations between specific styles and their chemical composition. HS-SPME-GC-MS has proven to be a useful tool as an approximation of human olfaction.

## Figures and Tables

**Figure 1 foods-12-01276-f001:**
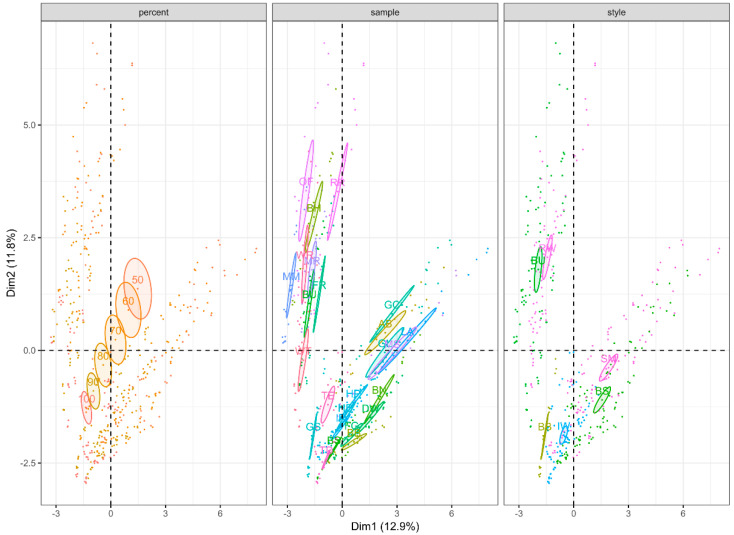
Multi factor analysis (MFA) of larger dataset of headspace volatile content of 25 whiskies consisting of five blended Scotch whiskies (BS), five Bourbons (BU), four Irish whiskeys (IR), four rye whiskeys (RW), six single malt Scotch whiskies (SM), one barrel strength Bourbon whiskey (BB), measured at six different dilution levels (100, 90, 80, 70, 60, and 50% whisky in purified water) in triplicate. Analysis was performed using nine groupings consisting of three categorical variables (percent whisky, style, and sample name) and 91 volatile compounds separated in to eight compound classes, tentatively identified by GC-MS using the NIST 2020 library. Ellipses represent a 95% confidence interval about the centroid.

**Figure 2 foods-12-01276-f002:**
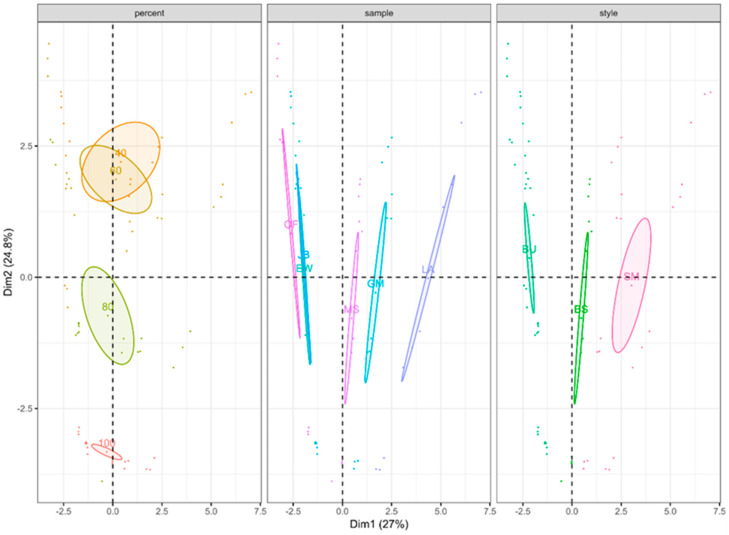
Multi factor analysis (MFA) of headspace volatile content of a subset of six whiskies used for descriptive sensory analysis (DA) consisting of three Bourbons (BU), two single malt Scotch whiskies (SM), and one blended Scotch whisky (BS). Headspace content was measured at four different dilution levels (100, 80, 60, and 40% whisky in purified water) in triplicate. Analysis was performed using 12 groupings, including three categorical variables and 131 volatile compounds tentatively identified by GC-MS using the NIST 2020 library, grouped in to 11 separate compound classes. Ellipses represent 95% confidence intervals about the centroid.

**Figure 3 foods-12-01276-f003:**
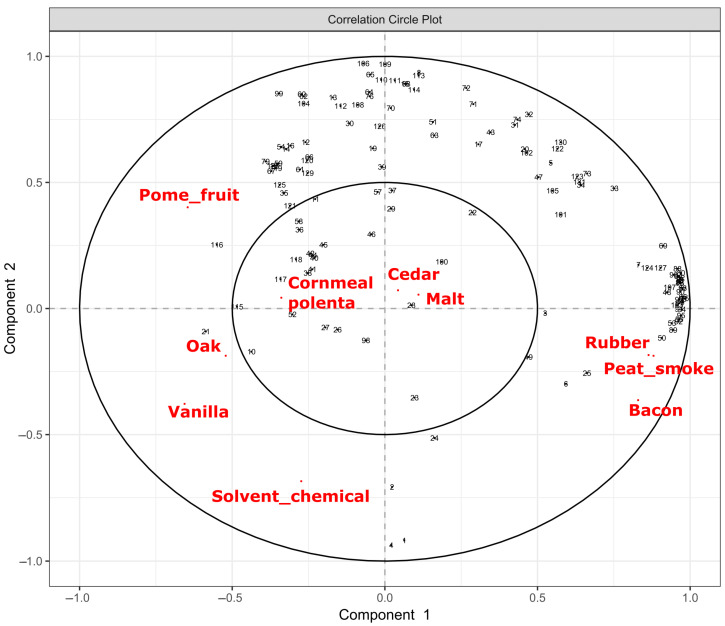
Partial least squares (PLS) variables plot of sensory attributes and their associated chemical compounds. A total of seven sensory attributes were found to be statistically significant (“Solvent/Chemical”, “Vanilla”, “Oak”, “Pome Fruit”, “Peat Smoke”, “Bacon”, and “Rubber”) while three were not (“Cedar”, “Malt”, and “Cornmeal/Polenta”).

**Figure 4 foods-12-01276-f004:**
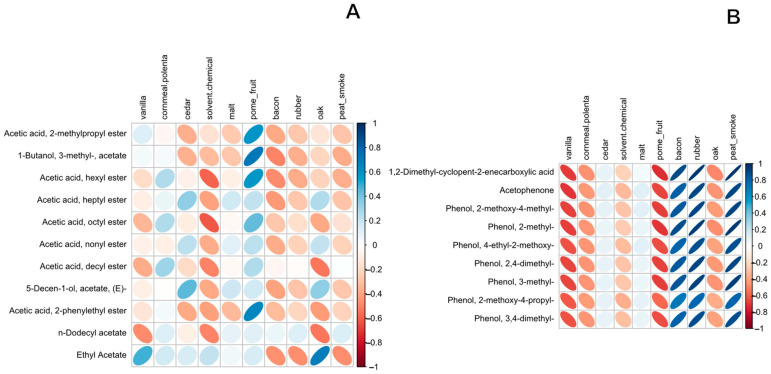
Correlograms of “acetate ester” (**A**) and “phenol” (**B**) compound classes with sensory attributes. Irregularity of ellipsis shape corresponds to the correlation coefficient while color represents positive/negative correlation (red = negatively correlated, blue = positively correlated). Volatile compound data were measured in triplicate by HS-SPME-GC-MS and tentatively identified using the NIST 2020 library, triplicate measures were averaged for comparison. Sensory attributes are averaged duplicate measures recorded by trained descriptive analysis (DA) sensory panel (*n* = 20).

**Figure 5 foods-12-01276-f005:**
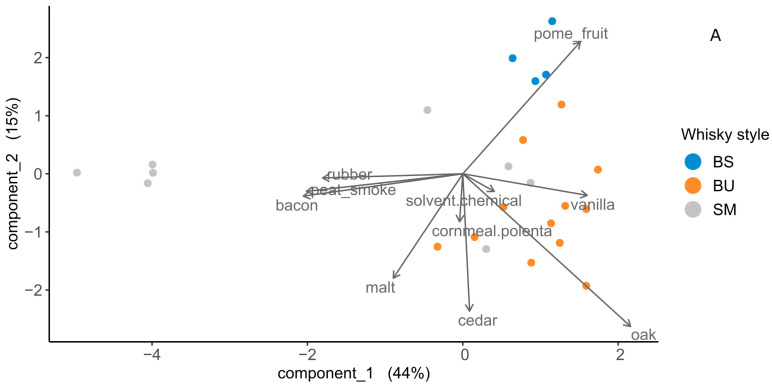
Partial least squares discriminant analysis (PLS-DA) biplot of sensory attributes and whisky styles (**A**) and dilution factor (**B**). In general, smokey terms (“Rubber”, “Bacon”, and “Peat Smoke”) were associated with peated single malt Scotch whisky (SM), “Vanilla” and “Oak” were associated with Bourbons (BU), and “Pome fruit” was associated with unpeated, blended Scotch whisky (BS). Undiluted whisky more directly correlated with “Solvent/Chemical”, “Vanilla”, “Bacon”, and “Oak” aromas while heavier dilution favored “Pome fruit”, and “Cedar” attributes.

## Data Availability

Data files and applicable R scripts can be accessed by contacting the corresponding author.

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
