# Peer review of "Impact of Dilution on Whisky Aroma: A Sensory and Volatile Composition Analysis"

_foods, 2023, doi:10.3390/foods12061276_

Round 1
Reviewer 1 Report
1. Why there are no mention of multifactor analysis, PLS and correlograms performed in this study in the abstract?
2. “volatile composition” keyword should be included.
3. The GC column dimension should be presented as “30 m x 0.250 mm ID, 0.25 mm film thickness” under section 2.3.
4. The volatile profiling by GC in terms of GC chromatogram and analysis data should be provided in the main text.
5. All the labels in Table 1-3, 5, 6 should be enhanced for clarity and visibility.
6. Some more discussion elucidating the proximity of data in the biplots shown in Figure 5 and 6. There are only 3-4 lines as discussion.
7. Figure 5 and Figure 6 should be combined as A and B parts of a single figure with a common caption.
8. Some discussion by comparing the data and/or results with that of similar type of reported papers dealing with impact on dilution of aroma beverages.
Reviewer 2 Report
Title: " Impact of dilution on whisky aroma: a sensory and volatile composition analysis ".
This paper provides important insight into how dilution of whisky during consumption changes consumer perception. It also shows the usefulness of HS-SPME-GC/MS as a proxy for human olfaction, which can be used to evaluate complex relationships between aroma and dilution in whiskies. The findings from this research could help distillers better understand their products’ flavor profiles at different levels of water concentration, allowing them to make more informed decisions about product development or marketing strategies.
I think it is an interesting work. However, it still requires a few improvements which are listed as below.
1. Lines 103-104, why did you choose to add 10 ml of solution to the 20 ml sample bottle? Please give a reasonable explanation. We know that the gas-to-liquid volume ratio between the volume of solution in the sample bottle and the gas above the liquid in the bottle is highly dependent on the aroma release. At thermodynamic equilibrium, different gas-liquid volume ratios correspond to different partition coefficients.
2. Line 110, to investigate the release of volatile compounds in the headspace, SPME is a convenient method, However, for the solution with small dilution ratio, that is, high alcohol content, it may be difficult to overcome the overload of fiber head on ethanol, which has a great impact on the results.
3. Line 116, why does this experimental analytical method require a split? Is it still representative for the detection of some trace compounds?
4. Line 164, what is the concentration and intensity of the corresponding standard reference when evaluating the intensity of these ten aroma attributes?l
5. Line 222, the total sum of the two principal components derived from the experimental results is only 24.7%, which is not sufficiently representative.
6. Line 260, the figure 3 looks terrible, not clear and concise, suggesting that the name of the compound could be adjusted to a code or something else.
Round 2
Reviewer 1 Report
The authors have satisfactorily addressed all the comments raised by reviewers and therefore I recommend acceptance of this article for publication.